# 3D Ultrasonic Brain Imaging with Deep Learning Based on Fully Convolutional Networks

**DOI:** 10.3390/s23198341

**Published:** 2023-10-09

**Authors:** Jiahao Ren, Xiaocen Wang, Chang Liu, He Sun, Junkai Tong, Min Lin, Jian Li, Lin Liang, Feng Yin, Mengying Xie, Yang Liu

**Affiliations:** 1State Key Laboratory of Precision Measuring Technology and Instruments, Tianjin University, Tianjin 300072, China; renjiahao@tju.edu.cn (J.R.); 1020202082@tju.edu.cn (X.W.); liuchang1311@163.com (C.L.); sunhe_ultrasonics@tju.edu.cn (H.S.); jktphy@tju.edu.cn (J.T.); tjupipe@tju.edu.cn (J.L.); 2Department of Mechanical Engineering, University of Wyoming, Laramie, WY 82071, USA; mlin4@uwyo.edu; 3Schlumberger-Doll Research, Cambridge, MA 02139, USA; lliang@slb.com; 4State Key Laboratory of Bioelectronics, School of Biological Science and Medical Engineering, Southeast University, Nanjing 210096, China; fyin88@gmail.com; 5International Institute for Innovative Design and Intelligent Manufacturing of Tianjin University in Zhejiang, Shaoxing 330100, China

**Keywords:** ultrasound, brain image reconstruction, machine learning, real-time imaging

## Abstract

Compared to magnetic resonance imaging (MRI) and X-ray computed tomography (CT), ultrasound imaging is safer, faster, and more widely applicable. However, the use of conventional ultrasound in transcranial brain imaging for adults is predominantly hindered by the high acoustic impedance contrast between the skull and soft tissue. This study introduces a 3D AI algorithm, Brain Imaging Full Convolution Network (BIFCN), combining waveform modeling and deep learning for precise brain ultrasound reconstruction. We constructed a network comprising one input layer, four convolution layers, and one pooling layer to train our algorithm. In the simulation experiment, the Pearson correlation coefficient between the reconstructed and true images was exceptionally high. In the laboratory, the results showed a slightly lower but still impressive coincidence degree for 3D reconstruction, with pure water serving as the initial model and no prior information required. The 3D network can be trained in 8 h, and 10 samples can be reconstructed in just 12.67 s. The proposed 3D BIFCN algorithm provides a highly accurate and efficient solution for mapping wavefield frequency domain data to 3D brain models, enabling fast and precise brain tissue imaging. Moreover, the frequency shift phenomenon of blood may become a hallmark of BIFCN learning, offering valuable quantitative information for whole-brain blood imaging.

## 1. Introduction

Advancements in medical imaging technology have ushered in an era where a comprehensive examination of human organs is now possible. Rapid and precise brain imaging allows for early diagnosis and evaluation of brain-related illnesses, as well as the establishment of therapeutic regimens. Despite the widespread use of MRI and conventional CT due to their high resolution (MRI: ~0.1 mm [1]; CT: ~0.23 mm [2]) [3], MRI is not suitable for people who are obese or have magnetic foreign bodies [4], and CT involves exposure to harmful ionizing radiation [5,6]. Additionally, these methods require bulky and expensive equipment and must be operated by healthcare professionals [7,8]. Furthermore, conventional B-mode is employed for imaging of the heart, abdomen, urinary, and digestive systems [9]. Nevertheless, the differing sound velocities and densities between the skull and soft tissues give rise to complications, such as reflection, refraction, and scattering of propagating ultrasound waves, resulting in complex and distorted intracranial wavefields [10,11]. The strong and high-amplitude reflections from the skull drown out the tiny pulses reflected from soft tissues, making high-resolution transcranial brain imaging extremely difficult [12,13].

One feasible solution to mitigate wavefield distortion is to utilize transcranial functional ultrasound by exciting and receiving signals through the open fontanelles of the skull [14,15]. However, it is essential to note that this technique is exclusively applicable to infants as the open fontanelles close as individuals age, and the angle dependence of the Doppler effect may limit the imaging angle [16]. Alternatively, intracranial signals can be acquired by puncturing the skull with a photoacoustic probe [17,18,19]. Nevertheless, compared to in vitro imaging, invasive methods present challenges in terms of operation and raise ethical concerns.

One recent solution has been to improve and design new tomography technologies. In this regard, photoacoustic imaging (PAI), ultrasonic imaging, and full waveform inversion (FWI) technology are promising solutions. Over the past 20 years, PAI has advanced in vascular disease, microcirculation, oncology, gastrointestinal disease, and brain function [20]. Li et al. used independent monopulse panoramic photoacoustic computed tomography to achieve real-time imaging of system dynamics in small animals and obtain anatomical and functional details of organs [21]. By scanning the entire breast in a single breath-holding session, Lin et al. determined whether the breast contained a healthy state of the tumor without resorting to ionizing radiation or exogenous contrast agents [22]. Shuai et al. used a hemispherical arrangement of ultrasound transducers around the human head to achieve functional imaging of hemicerebrectomy patients, particularly in infancy, with imaging results comparable to current standards for imaging human cerebral vessels and function [23]. Patrick et al. demonstrated non-invasive whole brain imaging of tau targeting PBB5 probes using volume multispectral photoacoustic tomography (vMSOT) in a 130 m resolution 4-replicate tau P301L model [24]. PAI combines high optical contrast with high ultrasonic resolution, but its maximum detection depth in breast tissue (about 106.4 mm [25]) remains a limitation for wider applications.

Ultrasound imaging encompasses B-mode, M-mode, Doppler imaging, contrast imaging, and ultrasound elastic imaging. B-mode ultrasound utilizes multi-channel array configurations such as linear, convex, and 2D arrays, enabling the provision of 2D/3D information about anatomical or pathological structures and dynamic changes in organs. M-mode ultrasound, while offering a slower scanning speed than B-mode ultrasound, is better suited for repetitive scanning of specific tissues or organs. Researchers have developed a novel ultra-fast contrast-enhanced ultrasound Doppler technique to monitor the structure and function of spinal vessels in real time by establishing an ultrasonic experimental platform combined with a 15 MHz linear array transducer [26]. Other studies have explored the effect of pulse length and concentration on phase-change contrast agents vaporization signals to form high contrast in vivo ratio images of tissue [27]. These techniques have led to various ultrasound molecular imaging methods and the ongoing development of clinical molecular contrast media. For example, Correia et al. proposed an ultrasound-based volumetric 3D elastic tensor imaging that allows quantitative volumetric mapping of the elastic properties of tissues in weakly elastic anisotropic media [28]. Additionally, Maresca et al. developed a cross-amplitude modulation technique for highly specific in vivo imaging of the gastrointestinal tract [29]. Forsberg et al. demonstrated that contrast-enhanced ultrasound can non-invasively measure neovascularization of glioma and breast tumors in rat xenograft models [30]. Clinical contrast-enhanced ultrasound (CEUS) imaging and intravascular microbubble tracking algorithms can be used to draw cerebral blood flow maps in a pediatric porcine model of hydrocephalus [31]. These methods have been widely performed on animal models, newborn children, and adults in preclinical studies. While these methods have demonstrated great potential in the diagnosis and treatment of brain diseases, they have not been applied to high-resolution 3D real-time imaging of the entire brain due to their limitations.

The FWI is an imaging technology extensively used in the geophysics community and recently introduced into medical imaging [32]. Guasch et al. successfully generated accurate 2D and 3D brain tomography images with sub-millimeter resolution using wavefield simulation data based on FWI [33]. To address the problem of costly forward problems in FWI and uncertainty in the assessment of brain phantom image quality, Bates et al. proposed stochastic variational inference [34]. However, the iterative solving process of FWI in the Devito toolkit consumes a lot of computational resources and time due to the need for the second-order gradient information of the objective function (Hessian matrix) [35]. As a result, FWI is hindered in its application for clinical real-time medical imaging, as it can take up to 32 h for 3D imaging [33]. To solve the time-consuming problem of FWI inversion, Yue et al. constructed a novel deep convolutional neural network with the encoder–decoder structure to learn the regression relationship from a seismic waveform dataset to an underground model, achieving faster imaging times of 5.5 milliseconds for 2D imaging [36], while a convolutional neural network (CNN) architecture was proposed to automatically infer missing low frequencies without pre-processing and post-processing steps by considering bandwidth expansion as a regression problem in machine learning [37]. Tong et al. extended the fast inversion tomography to real-life 3D medical imaging problems based on the optimization framework of FIT, proposing a linear residual decomposition technique [38]. Although trained using synthetic data, the deep learning-driven FWI is expected to perform well when evaluated using sufficient real-world data. Despite the above works successfully applying network structures to physical FWI to solve the practical problem of inversion, our approach is different. We use network structures to explore the mapping relationship between wavefield frequency domain data and 3D brain models, avoiding redundancy by not using time domain data and reducing the amount of data to be processed to a certain extent. Moreover, we obtain 3D brain images at one time without the need for the superposition of 2D tomography images.

There is currently no universally applicable and harmless technology for 3D high-resolution real-time imaging of the human brain. In this paper, we propose a deep learning framework based on full-waveform forward modeling as pilot research in medical imaging. The paper is structured as follows: Section 2 introduces the wavefield data acquisition method and the architecture of the BIFCN, starting with the theory of 3D wave equations. Section 3 and Section 4 present the numerical simulation and laboratory validation results, including the adjustment and optimization of the number and size of convolutional kernels, pool layer size, and multiparametric multiscale fusion of the BIFCN for optimal imaging results. We demonstrate the feasibility of BIFCNs for 3D brain imaging using experimental acoustic data in the laboratory. Finally, we provide a discussion and summary of our findings in Section 5 and Section 6.

## 2. Methods

### 2.1. Three-Dimensional Wavefield Forward and Modeling

Forward simulation refers to calculating the theoretical wavefield distribution described by a wave equation, which is constructed by a mathematical and physical model that takes into account the properties of the medium. Considering the physical characteristics of biological tissue, the 3D acoustic wave equation for isotropic media is used to describe ultrasonic wave propagation in the human brain [39], which is defined as
(1)$$\rho (r)\frac{\partial }{\partial r}\left(\frac{1}{\rho (r)}\frac{\partial p(r,t)}{\partial r}\right)=\frac{1}{c{\left(r\right)}^{2}}\frac{\partial^{2}p(r,t)}{\partial t^{2}},$$
where p(r,t) is the pressure wavefield at point r at time t; ρ(r) and c(r) are the density and velocity at point r. To obtain the acoustic wave equation in the space–frequency domain, the temporal Fourier transform can be applied, which yields
(2)$$\left(\frac{\partial^{2}}{\partial r^{2}}+{\left(k_{r}\right)}^{2}\right)\psi =-O(r)\psi ,$$
with $k_{r}=2\pi f/c_{r}$ denoting the wavenumber of the background at point r. f is the frequency and $c_{r}$ is the velocity of the background at point r, ψ is the vector expression of the wavefield, equal to the Fourier transform of the pressure field. O(r) is the mathematical expression of the scatterer and is defined as [40]
(3)$$O(r)=k_{r}^{2}\left[{\left(\frac{c_{r}}{c(r)}\right)}^{2}-1\right]-\sqrt{\rho (r)}\frac{\partial^{2}}{\partial r^{2}}\left(\frac{1}{\sqrt{\rho (r)}}\right).$$

We utilize the clot as the disturbance term in the context of wavefield forward modeling. The primary objective of the paper is to explore and understand the relationship between brain models with different clots and wavefield disturbance. Thus, Equations (4) and (5) denote the background wavefield and disturbance wavefield, respectively, which are defined as [41]
(4)$$\left(\frac{\partial^{2}}{\partial r^{2}}+k_{r}^{2}\right)\psi_{r}=0,$$
(5)$$\left(\frac{\partial^{2}}{\partial r^{2}}+k_{r}^{2}\right)\zeta_{r}=\delta ,$$
where δ is Dirac delta. $\zeta_{r}=\exp(ik_{r}R)/4\pi R$ can be solved analytically in 3D space, and R is the distance between the source and measurement point.

The finite-difference method is utilized to calculate the wavefields of the time domain based on Equations (2)–(5) and the physical model of the medium.

Subsequently, a numerical simulation database is established as the data labeled to train the network [42]. Figure 1 depicts the geometry of the wavefield forward modeling and the simulation. In this study, we employ a model derived from the MIDA model developed by Iacono et al. [43]. As illustrated in Figure 1a, sensors are placed around the brain, and the 3D physical model is converted into the acoustic model based on the physical acoustic dataset developed by Batalla et al. [44]. The simulated wavefield snapshots at different times are presented in Figure 1b. All transducers are employed to record the wave signals around the brain model at each excitation. It is important to note that our approach in this study does not involve the use of an ultrasonic focusing transducer array or algorithms for 3D imaging in subsequent experiments, which is different from traditional transcranial ultrasound.

### 2.2. Two-Dimensional Fully Convolutional Network

The purpose of the network structure is to obtain the 3D brain-clot map by processing the time–frequency domain wavefield data. An FCN solves the problem of semantic-level image segmentation for the end-to-end pixel-level prediction of images by converting the fully connected layer in CNN into multilayer convolutional layers. The input of the traditional FCN network can be a color image of any size, and the output is the same size as the input. However, the input and output of a traditional FCN network are not suitable for handling the mapping between the 3D space model and data. In this paper, we modify the input to a digital matrix containing wavefield information and adjust the output to a 3D brain model with a size defined according to the desired resolution. This modification is shown in the red box in Figure 2a. It is noteworthy that the selection of the network architecture underwent multiple iterations, with various alternative network structures proving entirely ineffective in terms of inversion performance. This fact constitutes one of the primary reasons for our endorsement of the network depicted in Figure 2a as the optimal choice. Additionally, the optimization of network hyperparameters fell within the scope of our considerations.

To optimize the relationship between the perceptual field, network parameters, and convolutional kernels, it is advisable to start with a small initial convolutional kernel size while maintaining a sufficient perceptual field. As the number of layers increases, gradually decreasing the size of the convolutional kernel and pooling filter can help obtain more details and approach the size of the network output. Meanwhile, the number of convolutional kernels should always be consistent with the size of the output 3D model. As the number of convolutional layers increases and the size of convolutional kernels gradually increases, the corresponding training speed decreases. Finally, the last convolutional layer directly outputs the reconstruction images.

The performance and efficiency of training and testing can be affected by different network architectures. Deeper network architectures may enhance performance, but require longer training time. On the other hand, shallow network architectures have better computational efficiency but can lead to underfitting of the network structure and ultimately unsatisfactory results [45]. Therefore, choosing a suitable network structure is important for imaging performance. In this paper, the BIFCN network structure is used for simulated brain imaging with different depths. It includes four convolutional layers, one pooling layer, and a variety of convolutional kernels.

The wavefield data are processed by the network model as follows: (1) The signal data from the sensors undergo a Fourier transform, converting it to the frequency domain. (2) The optimized frequency amplitudes are then extracted to create a new two-dimensional matrix. All the matrices are normalized to form the input layer matrix. The convolutional layer is connected after the input layer to extract features from the frequency domain normalized data. It is mainly responsible for feature learning and contains several feature mappings processed by the convolution kernel. Each convolutional kernel processes only the received domain data, using the same shared weights, which reduces the number of free parameters and allows the BIFCN to perform deeper processing with fewer parameters. The formula for calculating the convolution layer is shown below.
(6)$$Conv(m.n)=R\left(\sum_{u=1}^{L-1}\sum_{v=1}^{L-1}w_{u,v}S_{m+u,n+v}+B\right),$$
where Conv(m.n) represents the convolution result, R represents the activation function, $w_{u,v}$ represents the weight of the convolution kernel on row v and column of u, S is the input layer, and B is the bias. The convolution process enables the feature extraction of the input layer image.

The leaky Rectified Linear Unit (ReLU) activation function is introduced after each layer of convolution operation to prevent the neural network from behaving like a linear regression model and to solve the dead ReLU problem, i.e., the phenomenon that some neurons cannot be activated due to improper parameter initialization.

Since the adjacent units within the 2D network share identical values, the convolutional layer’s adjacent outputs have similar values as well. This similarity could potentially lead to overfitting, redundant network structural parameters, and computational waste. To tackle this challenge, downsampling and downscaling the feature maps are used to represent the large feature maps with small feature maps. This reduces computational effort and simplifies the network complexity, improving the generality of the network structure. Pooling is the specific implementation process used for this purpose, as shown in Figure 2b.

The output size of the model is x = 240 mm, y = 200 mm, and z = 100 mm and the grid spacing is 1 mm. The output layer is a convolutional layer, in which 240 and 200 represent the size of the 2D slice, and 100 represents the number of slices or convolutional kernels. The network structure is trained by optimizing the network parameters to achieve the minimum loss function, and the loss function is defined as the mean square error (MSE) between the true velocity map and the reconstructed velocity map.
(7)$$MSE({\vec{v}}_{p},{\vec{r}}_{p})=\frac{1}{n}{\sum_{p=1}^{n}\left\|\left[{\vec{v}}_{p}\right]-\left[{\vec{r}}_{p}\right]\right\|}^{2},$$
where $v_{p}$ represents the horizontal cross-section plane velocity diagram, and $r_{p}$ represents the output of BIFCN.

The neural network was trained using the adaptive moment estimation (Adam) algorithm, an optimization algorithm that relies on low-order moments. This algorithm is a first-order gradient stochastic objective function optimization method that adjusts the learning rate of each parameter iteration to ensure that parameter changes are relatively smooth within a predefined range. In comparison to the Root Mean Square Propagation algorithm, Adam demonstrates superior convergence in network training, requires less computer storage capacity, and allows for constant diagonal rescaling of the gradient, which makes it suitable for solving problems with large datasets and multiple parameters [46]. The velocity map predicted by the network for the 3D brain model best matches the true velocity map when minimizing the L2 norm in Equation (7).

### 2.3. Dataset Preprocessing and Reconstructed Image Evaluation

#### 2.3.1. Dataset Preprocessing

Within each group, the maximum amplitude of the frequency domain data fluctuates depending on the clot’s size and location. To maintain the frequency domain amplitude as the sole variable during the reconstruction process, it becomes imperative to preprocess the network input through amplitude normalization. This preprocessing step yields a numerical matrix sized at 512 × 512, serving as the network’s input for subsequent analysis.

#### 2.3.2. Image Correlation Coefficient

In the paper, the quality of the reconstructed velocity maps needs to be evaluated. Pearson correlation (PC) was used to evaluate the accuracy of sample reconstruction [47]. The calculation formula of PC is shown as
(8)$$\mathrm{PC}=\frac{\sum_{i=1}^{n}\left(X_{i}-\bar{X})(Y_{i}-\bar{Y}\right)}{\sqrt{\sum_{i=1}^{n}{\left(X_{i}-\bar{X}\right)}^{2}}\sqrt{\sum_{i=1}^{n}{\left(Y_{i}-\bar{Y}\right)}^{2}}},$$
where X and Y represent the true and the reconstruction imaging of the samples, respectively.

## 3. Numerical Simulation

In the numerical experiments, uniform round (2D) or spherical (3D) clots with different sizes and positions are generated in the human brain, and sensor arrays with the same shape as the brain around it are placed. During the signal acquisition process, one sensor is used as the excitation source, and all sensors act as the receiving source. This process is repeated 512 times until all sensor units are used as excitation sources, at which point a complete signal collection of a clot in the brain is completed. The time domain signals are converted into the frequency domain and the amplitude of characteristic frequency is extracted to construct a feature matrix. The network structure is optimized to recognize the mapping relationship between the brain model and the frequency domain matrix and the trained network is then used to process the frequency domain matrix to reconstruct the high-precision brain model. The network operates with frequency domain data matrices as its input and produces two-dimensional or three-dimensional brain models as its output. This approach simplifies and streamlines the inversion process when compared to traditional methods. During training, the necessity for the iterative and computationally intensive finite-difference time domain wavefield simulations and inversion model iterations is eliminated. Instead, the network directly seeks the nonlinear mapping relationship between the frequency domain data matrices and the resulting brain model. Meanwhile, to eliminate the residual wavefield information from the previous excitation source in the model, a set of data acquisition intervals should be greater than 0.2 ms, taking into account the model size and sound velocity distribution.

We utilized the MIDA brain model introduced in Section 2 and solved the wavefield equations while varying the size and location of the target clot within the skull. This allowed us to establish a wavefield dataset containing information on reflection, refraction, and scattering. Figure 3a shows the size and relative position of a sample of target clots used in the simulation and the brain model. Figure 3b presents a set of frequency domain data matrices corresponding to the model. It is noteworthy that these frequency domain data matrices exhibit highly distinctive features. The amplitude is maximal in the self-transmit and self-receive signals of the sensor unit, gradually diminishing with increasing sensor distance. The shape of the frequency domain data matrices closely correlates with the sensor positions and their respective identifiers. Our wavefield modeling was performed in the finite difference simulation environment using the Python-based Devito toolkit [48], which includes 2D and 3D models. All sample programs were run on 16 Intel Ivy Bridge CPUs. For the simulation, the model was divided into 480 × 440 × 200 grids with 0.5 mm × 0.5 mm × 0.5 mm grid spacing.

The zero-phase Ricker wavelet signal was used as the excitation source, with an excitation frequency of 700 kHz [33], which was selected based on a balance between penetration efficiency and resolution. The excitation waveform and its frequency content are shown in Figure 4. The size and location of the clot in the brain model were randomly generated for each realization, with the radius set to be between 9 and 15 mm according to National Institutes of Health statistics [49]. Meanwhile, the clots were set to either circular (2D) or spherical (3D) with a constant velocity distribution of 1700 m/s [43].

To optimize performance based on hardware resources, the batch size, the total epoch, and the early stopping epoch are respectively specified as 8, 5000, and 500. This means the training termination condition is either when the program training epoch reaches 5000 or when the training loss function results stop decreasing within 500 epochs. The optimal network achieved is either the model with the lowest MSE that does not change within the early stopping epoch or the last model that reaches the total epoch. The training and validation sets are randomly divided during the training process to improve the robustness and generalization of the network structure [50].

### 3.1. Two-Dimensional Horizontal Cross-Section Simulation Experiment

In the 2D horizontal cross-section simulation, we utilized a model size of 250 × 250 and a grid interval of 1 mm. To capture the variability in the data, we generated a speed model database consisting of 700 samples, with a ratio of 0.6:0.2:0.2 for training, validation, and test sets, respectively. The training dataset consists of 420 samples, the validation dataset comprises 140 samples, and the testing dataset contains 140 samples. During the training process, we employed 420 training samples and 140 validation samples to establish the mapping relationship between the two-dimensional brain models and data. Throughout the training, we kept the hyperparameters unchanged, ensuring a consistent learning environment. The initial weight of the BIFCN network was set to a random number between 0 and 1. Following the training phase, we tested the network structure online using 140 test sets to evaluate its effectiveness. Remarkably, we could perform whole-brain reconstruction without an initial model, thereby avoiding reliance on an initial model, as is typically necessary for FWI.

The effectiveness of BIFCN in reconstructing the brain cross-section is vividly presented in Figure 5. Specifically, Figure 5a displays the ground truth velocity map of the test sample, while Figure 5b depicts the reconstructed results of BIFCN. Our results showcase the remarkable accuracy of BIFCN, with a phenomenal PC of 99.95332%. Further analysis of the extracted regions from Figure 5a,b reveals that the velocity fluctuation between the reconstructed and true velocity is minimal and stable. This remarkable stability indicates that BIFCN has great potential for accurately reconstructing complex brain models.

### 3.2. Three-Dimensional Brain Simulation Experiment

The 3D simulation utilized a high-resolution model consisting of 240 × 200 × 100 grid numbers with a 1 mm grid spacing. A total of 401 samples were included in the dataset, with a balanced allocation ratio of 0.6:0.2:0.2 between the training, validation, and test sets. The training dataset comprises 240 samples, the validation dataset consists of 81 samples, and the testing dataset includes 80 samples. Following the same procedure as in the 2D horizontal cross-section simulation, BIFCN was trained offline using 240 training samples and 81 verification samples, while 80 test samples were used for online evaluation of the network’s performance.

Figure 6 showcases the performance of the BIFCN by comparing the training and validation loss. As seen in the graph, the MSE of the network model decreases with training iteration, demonstrating a rapid decline in the beginning and a steady decrease towards the middle and end of training. It is noteworthy that the training set displays a more stable MSE variation trend with less value fluctuation than the validation set. Upon achieving the optimal model, the MSEs of the training and validation sets are 2.398 × 10^−4^ and 2.429 × 10^−4^, respectively. To further illustrate the imaging performance of the network model, Figure 7 depicts the results obtained for a randomly selected set of test data. Specifically, the 3D velocity map (Figure 7b) was recovered in specific sections using the data simulated from a selected model in the database, with the true model shown in Figure 7a. In addition, to evaluate the reconstruction quality of the clot more distinctly, Figure 7c,d exhibit one horizontal cross-section of the true and reconstruction results for comparison, respectively. The results demonstrate that the demarcation between the clot and the tissue exhibits a clear boundary with consistent size and location. Moreover, the 1D velocity profile images are delineated in the 2D cross-section, as highlighted by the red and blue lines in Figure 7e,f. The red and blue lines in Figure 7e,f represent the respective parts of Figure 7c,d. The black line corresponds to the true velocity, whereas the red dashed line denotes the inverted velocity in Figure 7e. Finally, the performance of the network model was quantified using the PC between the true image and the reconstruction result, which resulted in a high value of 99.61506%.

Figure 8 showcases the statistical findings of PC between the 3D reconstruction and the true images, revealing the exceptional proficiency of this network architecture. The average PC of the test samples is an outstanding 98.54041%, serving as a testament to the robustness and durability of the model.

The set of 80 test samples was categorized into three distinct groups based on clot size, and Table 1 exhibits the distribution of PC findings. The statistical outcomes elucidate that as the clot diameter decreases, the PC also experiences a decline, thereby causing a slight diminishment in the quality and resolution of the reconstructed image. Nonetheless, the impact of this reduction is negligible.

## 4. Laboratory Experimental Results

### 4.1. Experiment Preparation

To assess the practicality of BIFCN imaging in a true brain-clot phantom, an experimental device was developed as illustrated in Figure 9. The brain phantom and its design drawing, which were utilized for in vitro experiments with a resin material, are presented in Figure 9a,b, respectively. The design drawings are based on the actual human brain, omitting soft tissue and other internal structural components. The phantom is fabricated using state-of-the-art 3D printing technology. To replicate the characteristics of the skull, we harness the attributes of high density, and elevated sound velocity attained through the resin curing process. In simulating the acoustic impedance contrast between the skull and soft tissues, we leverage the acoustic impedance differential between the resin and water. It is noteworthy that during the fabrication process, careful material selection for the phantom is imperative to achieve a closer approximation of the acoustic parameters of the skull. The physical parameters of the resin material are provided in Table 2. The phantom boasts ultra-high spatial resolution and accuracy, with a machining precision of 0.3 mm. The clot phantom can be positioned anywhere within the brain phantom and connected via an aqueous medium. The velocities of the skull and clot phantom were measured to be 2618 m/s and 2222 m/s, respectively. Compared to conventional 2D and 2.5D phantoms, a 3D phantom offers distinct benefits, eliminating concerns about wavefield confusion resulting from out-of-plane effects in 2D assumptions.

The wavefield data were acquired utilizing a 512-channel transducer array, securely affixed around the phantom with rapidly polymerizable adhesive, based on ethyl cyanoacrylate, ensuring a close fit between the phantom and transducer, thereby resulting in higher signal-to-noise ratio data. The sensor unit, arranged in layers from top to bottom of the human brain, comprises a cylinder with a diameter of 5 mm, functioning as an acquisition unit, rather than a storage unit. The impedance measurement of the sensor unit is illustrated in Figure 9c. The center frequency of the sensor unit is 0.721 MHz, with a −3 dB bandwidth of 0.276 MHz. The electromechanical coupling coefficient is 0.577. The directivity is depicted in Figure 9d. The spacing between layers is 10 mm, with each layer of the sensor unit arranged at an equal angle, thereby facilitating the collection of a full range of wavefields across the brain and providing signals with more information about tissue distribution in the brain [31]. The 512-channel ultrasonic transducer is driven by the UTA-1024-MUX panel of the American Verasonics company (Kirkland, WA, USA), with the precise positioning of transducers to the phantom known. The excitation signal, a Ricker wavelet signal with 10 periods, is used to stimulate the transducers, with a center frequency of 700 kHz, and the excitation signal and frequency contents are illustrated in Figure 4a,b.

Although sub-MHz frequency signals have proven to be more readily penetrable through the brain and provide higher signal-to-noise ratios, they result in a compromise in imaging resolution. Figure 10a presents the true wavefield data recorded by 54 sensors in the set transducers array, distributed around the brain phantom from a single excitation source located in the yellow circle. Figure 10c,d show the time domain signal and frequency content recorded by the single receiver opposite the source, with the data displayed indicated by the yellow line. The signal possesses a good signal-to-noise ratio when penetrating the phantom, which aligns with experimental experience. The signal-to-noise ratio of the amplitude of the first arrival signal and the noise was computed, displaying 17 dB. The single acquisition process was repeated 512 times, and the time domain signal acquisition of one position was accomplished. The clot phantom is controlled by a 3D coordinate machine to move to a fixed position, with the wavefield collected to establish a database. Figure 10b illustrates the transducer section and the position of the wavefield shown in Figure 10a.

During the data acquisition process, the wavefield information obtained contains valuable insights into the multiple reflections, refraction, and scattering of acoustic waves. To ensure the accuracy of the acquired data, particular attention is paid to capturing the phase and amplitude changes of the ultrasonic signal resulting from the medium interface interaction, including boundary scattering, phase distortion, and amplitude attenuation. It is worth noting that waveform conversion attenuation due to the interface between the skull shell and the aqueous medium is also included. These details are crucial for subsequent reconstruction and must be captured during the in vitro experiments.

The comprehensive database was composed of 96 samples, which amounted to a staggering 49 gigabytes of complete wavefield data obtained from 50 mm diameter spheroid phantoms. The relative positions of the clot and brain phantom were meticulously controlled using a high-precision 3D displacement coordinate machine model XYZM178H-400D, ensuring the accuracy and consistency of the acquired data. While the spheroid phantom employed in this experiment offers a simplified representation of the true brain structure, it serves as a valid tool for verifying the feasibility of the proposed algorithm. However, for future investigations, more sophisticated brain phantoms with intricate structures should be utilized to further enhance the efficacy of this method.

During the training phase, a total of 57 training samples and 19 verification samples were employed to facilitate the offline training of the network structure. It is noteworthy that the hyperparameters were kept unchanged throughout the training process, in line with the simulation settings. Additionally, the initial weight of the network structure’s parameters was randomly set between 0 and 1. Once the training was accomplished, the performance of the network structure was evaluated by subjecting 19 test sets to online testing, and the reconstructed results were evaluated using the PC, a reliable measure of accuracy.

### 4.2. Reconstruction Results

Figure 11 showcases the imaging results obtained from both a training and test sample. The true and reconstructed 3D images are displayed in Figure 11a–d, which indicate that the size and position of the clot have been inferred accurately. To further verify the fidelity of the reconstructed velocity distribution within the clot, a two-dimensional profile of the model is presented in Figure 11e–h. Notably, the velocity distribution of the clot in the reconstructed images matches the design image with remarkable precision, resulting in a high PC of 97.9532% and 96.9871% for the 2D cross-sections of the reconstruction and true image, respectively. Moreover, the quality of the reconstructed images obtained from the in vitro experiments is on par with the numerical experiments in terms of the PC.

The fundamental principle of the algorithm is to establish a coherent mapping between the experimental observations and the 3D velocity model to generate a smooth transition at the clot’s boundary without any sudden velocity changes. The process is illustrated in Figure 11i,j, where slicing a 1D velocity profile from the 2D velocity distribution results in jagged variations in velocity within the clot. This is because it is challenging to identify a continuously differentiable function that precisely matches the theoretical model of uniform velocity distribution. The PC of the test sample between the reconstructed and true images on the 3D structure depicted in Figure 11 is 92.16076%, demonstrating the effectiveness of the proposed algorithm.

Figure 12a illustrates the training and validation losses of the experimental data on the intricate network architecture. The optimal models were identified for both the training and validation data, resulting in MSEs of 2.252 × 10^−4^ and 5.471 × 10^−4^, respectively. Although the network structure was subjected to some level of noise during experimentation, it managed to extract these minor features with negligible impact on the resulting images.

To evaluate the effectiveness of the proposed deep learning method for medical 3D imaging, a comparison was made between the reconstructed and true images of 20 testing and 19 validation samples. The statistical results are displayed in Figure 12b, which demonstrates a consistent overlap between the true and reconstructed images. The average PC obtained was 84.88228%, further highlighting the tremendous potential of the deep learning method for medical 3D imaging. In the allocation of the training, validation, and testing datasets, an absolute randomization principle was employed to minimize the influence of biases and randomness on experimental outcomes. Moreover, careful consideration was given to ensuring uniformity in the allocation, thereby guaranteeing that the positions of data points within the datasets were unrelated to any specific clustering. Consistency was maintained by applying the same allocation principles across multiple experiments. Subsequent random shuffling of the experimental models and reiterations yielded consistently favorable inversion results. The average PC for replicative experiments, conducted in accordance with meticulously documented experimental logs, encompassing critical details such as experimental conditions, instrument settings, and procedural steps, reached an impressive 84.5246%. Furthermore, with the combination of high-performance workstations and greater computational power, real-time high-resolution imaging of the true human 3D brain is now a feasible and exciting possibility, even with the current limitations in computational power available in the laboratory.

## 5. Discussion

### 5.1. Advantages and Achievements

In our laboratory, we developed and thoroughly characterized a remarkable 3D brain imaging reconstruction framework based on the BIFCN acoustic technique. Leveraging the 3D brain clot phantom, we have demonstrated the outstanding performance of our proposed approach. Notably, we replaced the fully connected layer in the FCN last layer with a convolutional layer to avoid excessive computational resources and time, thus ensuring the efficiency of the framework. Through extensive simulation experimentation, we have unequivocally shown that BIFCN can accurately locate the clot’s size and location while imaging brain tissues by effectively learning the 3D brain wavefield information. We employed randomly generated examples to train a series of models and wavefields, thereby revealing the sensitivity and reliability of BIFCN to the characteristic information of the wavefield. With the true brain model, BIFCN achieves reconstruction results with an average PC of 98.54041%, underscoring its unparalleled precision and accuracy. These findings open up new vistas of opportunities for the development of highly efficient and accurate brain imaging reconstruction frameworks.

Fast imaging is one of the hallmarks of the BIFCN framework, setting it apart from other brain imaging reconstruction techniques. The BIFCN process comprises offline training and online testing, and its remarkable efficiency is highlighted by the mere 12.67 s it took to test 10 models using a single RTX 2080Ti GPU (NVIDIA, Santa Clara, CA, USA). By leveraging advanced offline training, BIFCN can significantly reduce patients’ waiting times for clinical applications, thus providing them with prompt and accurate diagnostic information.

Three-dimensional volumetric image reconstruction algorithms are widely used in medical imaging, where 2D images are sliced and assembled from different locations and orientations to synthesize the final 3D image. While 2D reconstruction algorithms are a popular choice due to their reduced computational cost and ease of data acquisition, they are limited in their ability to account for the complex scattering and refraction of ultrasound signals outside of the imaging plane. This can result in phase deviation and distortion of the ultrasound signal, leading to lower-resolution imaging. Furthermore, successful 3D-FWI reconstruction requires an initial model that is as close to the true model as possible, which can be challenging to achieve and may lead to non-convergence during iterative reconstruction, ultimately resulting in failed reconstruction. However, the BIFCN algorithm overcomes these limitations by acquiring the full wavefield information of the 3D model, including reflection and scattering, resulting in high-resolution 3D imaging without prior knowledge or assumptions. This approach is more robust to poor models and adverse conditions, making it an ideal choice for challenging medical imaging scenarios.

### 5.2. Limitations and Solutions

Despite the success in the numerical and laboratory experiments, there is a limitation that cannot be avoided—the establishment of the database. This limitation affects the ability of the training model to generalize. Our current approach relies on an experimental database, and the use of both simulated and experimental data is not possible due to the mismatch between phase and amplitude in the two datasets. To improve the generalization ability of the model and obtain higher-resolution brain images, a large amount of true brain wavefield data with different structures is required to extend the database. Therefore, it is essential to address the mismatch between experimental and simulated data. As a result, we are investigating the calibration and unification of the experimental data and numerical modeling, which will be presented in a separate publication.

The clot models used in our simulations and experiments are regular-shaped models with uniform velocity distribution, which is not possible in clinical settings. We need to continue researching the validation of regular models and irregular-shaped models with Gaussian velocity distribution for network performance, which will be the focus of our future work. Additionally, we will continue to search for materials with physical parameters more similar to brain tissue, including sound velocity, sound attenuation, density, etc. This will be addressed in our next publication.

Furthermore, it is worth highlighting that the training and validation processes in this study were exclusively conducted on a single GPU. The inclusion of wavefield features to augment the dataset would significantly escalate the computational requirements. Nonetheless, given the exponential growth in computing power, the training of the network architecture can be readily expedited by employing more potent GPUs or implementing more efficient programming languages.

## 6. Conclusions

In this paper, we propose a highly efficient and high-resolution 3D brain imaging method, BIFCN. The method’s efficacy was first established through rigorous numerical experimentation, in which the 3D wave equation was solved to simulate the ultrasonic wavefield. To generate large datasets for brain models with known velocity and unknown clot properties, we trained a BIFCN to map from the modeled signal to the 3D model. During the imaging stage, the fixed frequency amplitude information was extracted, and a global normalized input layer was established to reduce computational effort and enhance imaging speed while eliminating the influence of amplitude on the imaging quality. The numerical experimental results demonstrate that the imaging statistics of BIFCN exhibit an impressive 3D average PC of 98.54041% between the true image and the reconstructed image. Further experimental validation was conducted in the laboratory, which convincingly illustrated BIFCN’s ability to accurately predict the location, size, and velocity of tumor models. The average 3D PC of the true and reconstructed images was 84.88228%, which underscores the immense potential and value of the method for clinical 3D real-time imaging. Additionally, the frequency-shift effect of brain blood on sound has the potential to serve as one of the salient features for BIFCN learning, thereby providing a novel and quantitative idea for whole-cranial cerebral blood flow imaging.

## Figures and Tables

**Figure 1 sensors-23-08341-f001:**
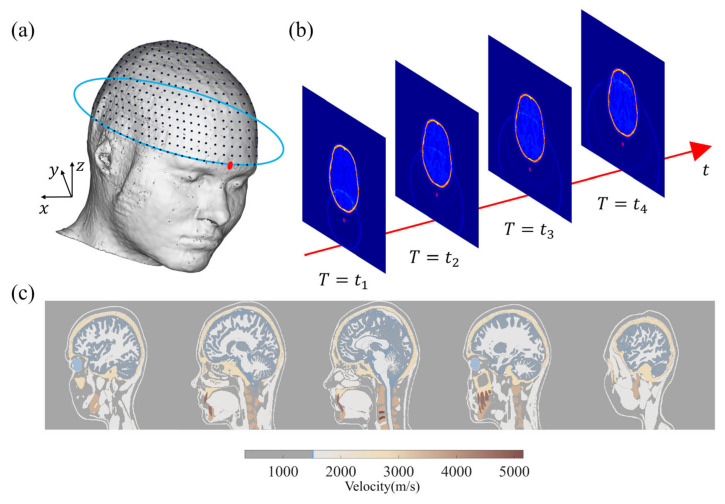
(**a**) MIDA model and sensor layout. (**b**) Two-dimensional display wavefield diagram of 3D craniocerebral wavefield forward modeling. (**c**) A 2D section of 3D acoustic model converted from MIDA model.

**Figure 2 sensors-23-08341-f002:**
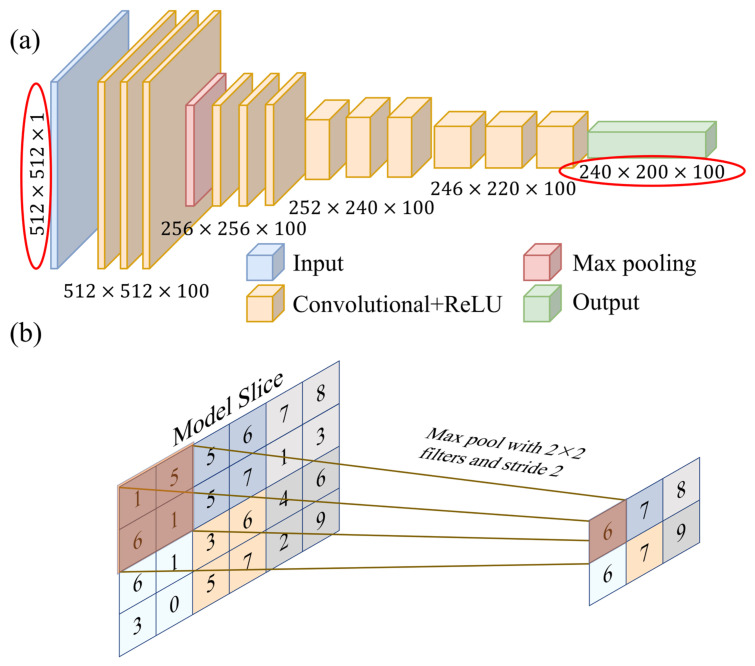
(**a**) Schematic of the BIFCN, including four convolutional layers and a maximum pooling layer. (**b**) Schematic diagram of maximum pool layer process.

**Figure 3 sensors-23-08341-f003:**
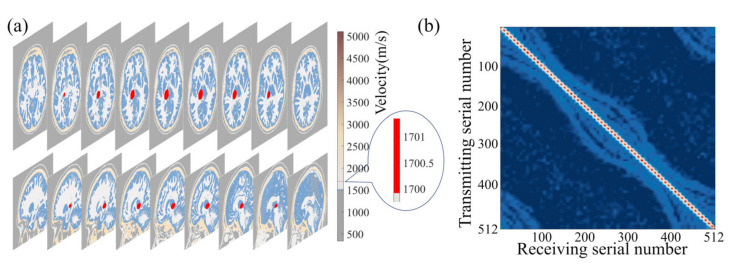
Schematic diagram of the relative size and position relationships of the 3D brain-clot model in numerical experiments.

**Figure 4 sensors-23-08341-f004:**
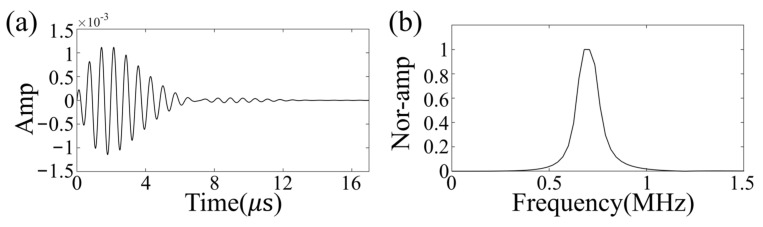
Excitation waveform (**a**) and its frequency content (**b**).

**Figure 5 sensors-23-08341-f005:**
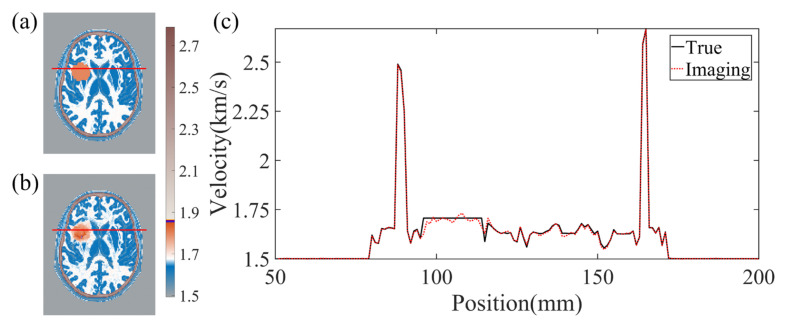
(**a**) The true velocity map of the test sample. (**b**) The reconstruction results of BIFCN. (**c**) is corresponding cross-section velocity at extracted position.

**Figure 6 sensors-23-08341-f006:**
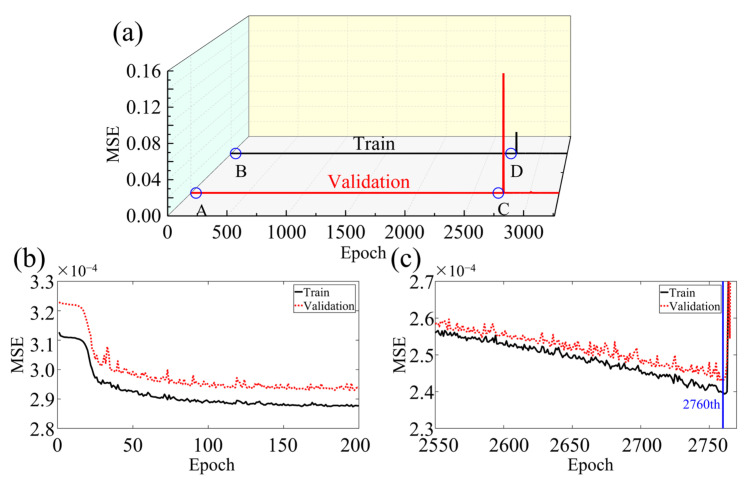
(**a**) Comparison of MSE results of training and verification sets in numerical experiments. (**b**) zooms in on A and B to show that the MSE curve decreases rapidly at the beginning of training. (**c**) zooms in on C and D to show that the MSE curve decreases rapidly at the optimal reconstruction result.

**Figure 7 sensors-23-08341-f007:**
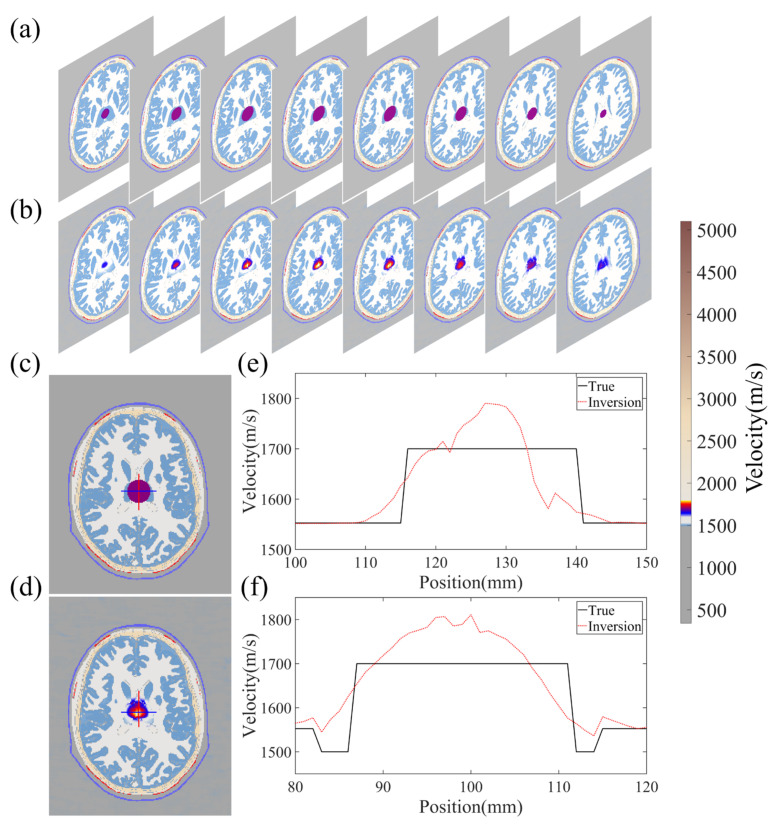
The reconstruction results of a random sample of validation models. (**a**,**b**) represent 2D cross-sectional views of the 3D model, (**c**,**d**) represent 2D frontal views of a certain cross-section, (**e**) indicates the velocity distribution values of the vertical red line part in (**c**,**d**), (**f**) indicates the velocity distribution values of the horizontal blue line part in (**c**,**d**).

**Figure 8 sensors-23-08341-f008:**
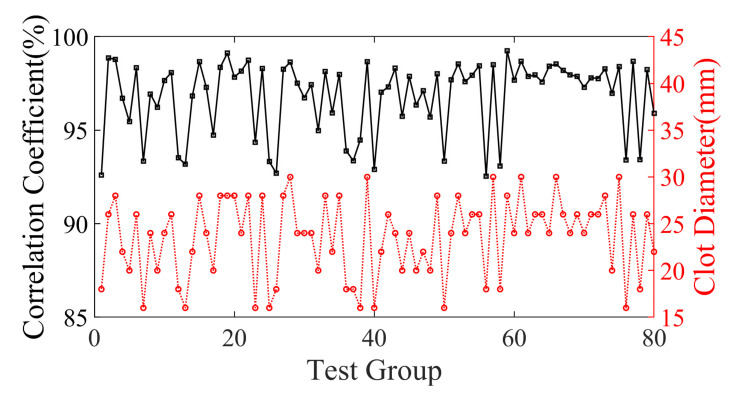
The black curve indicates the coefficient of overlap between the reconstruction results of the 80 samples of validation set models and the true models. The red curve indicates the corresponding clot diameter distribution.

**Figure 9 sensors-23-08341-f009:**
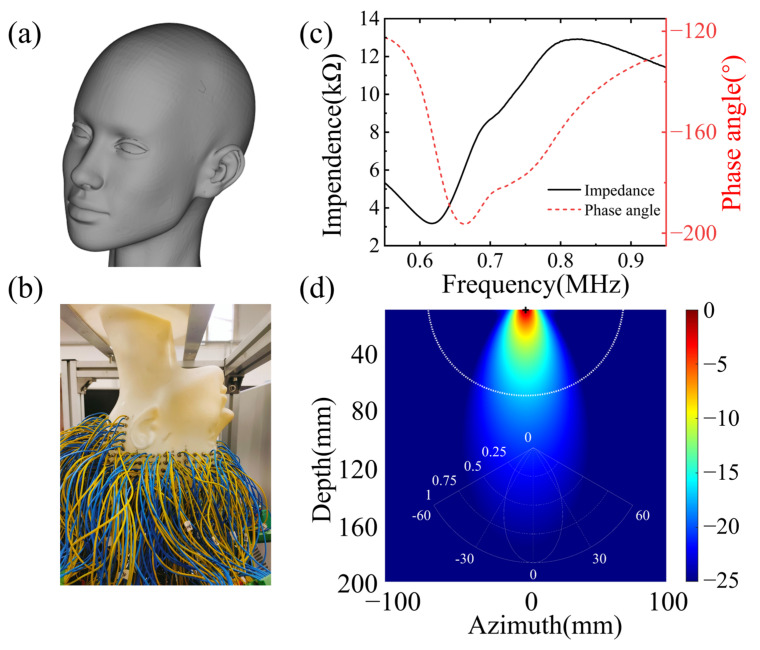
Experimental set-up. (**a**) Design model, (**b**) true model, (**c**) sensor unit impedance, (**d**) sensor unit transmitting beam.

**Figure 10 sensors-23-08341-f010:**
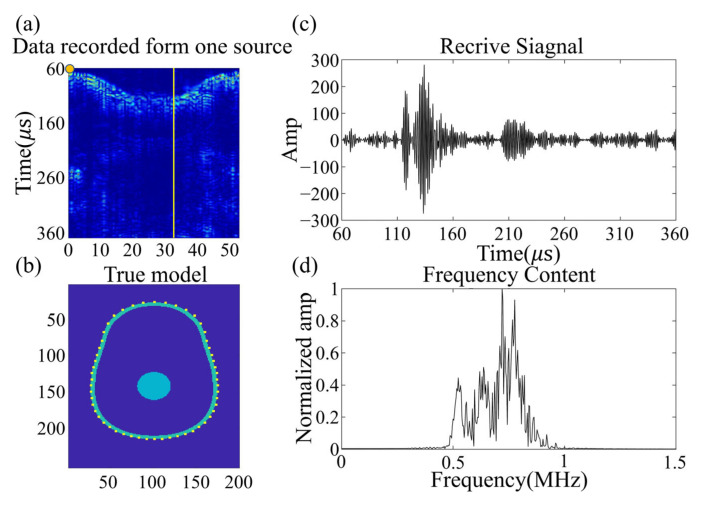
(**a**) shows the experimental data generated by a single source located at the yellow circle, recorded on an irregular elliptical array of 54 sensors around the head. (**b**) shows the shape of an irregular elliptical array. (**c**) displays the data recorded by a single receiver opposite the source; the position of the data shown is indicated by a yellow line. (**d**) displays the frequency spectrum of the received signal.

**Figure 11 sensors-23-08341-f011:**
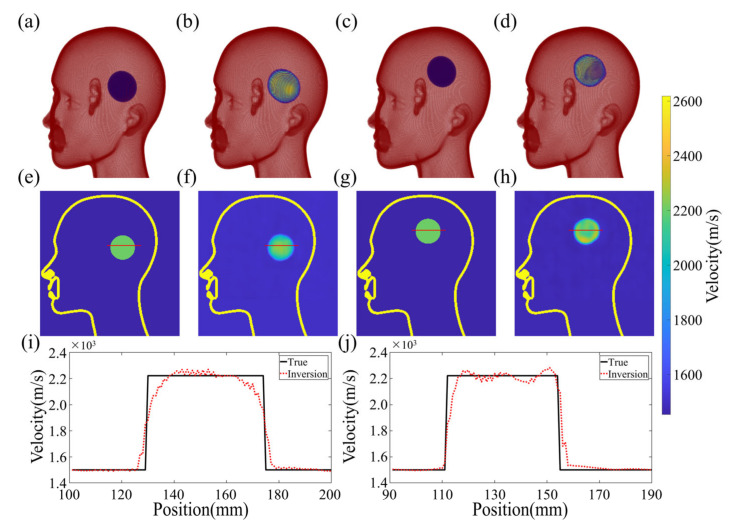
Reconstruction results of experiment data. (**a**,**b**) show the 3D results of true and reconstruction images in the training sample, respectively. (**c**,**d**) show the 3D results of true and reconstruction images in the validation sample, respectively. (**e**,**f**) respectively show the two-dimensional profile velocity distribution in the 3D model in the training sample. (**g**,**h**) respectively show the two-dimensional profile velocity distribution in the 3D model in the validation sample. (**i**) shows the one-dimensional profile velocity distribution of the model as shown in the (**e**,**f**) red line. (**j**) shows the one-dimensional profile velocity distribution of the model as shown in the (**g**,**h**) red line.

**Figure 12 sensors-23-08341-f012:**
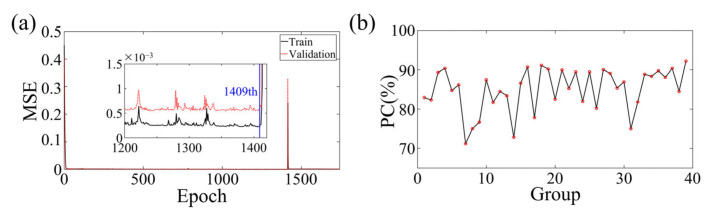
(**a**) The training and validation loss of BIFCN with the experiment dataset. The optimal models are saved at the 1409th epochs. (**b**) Statistical PC between true and reconstruction images of numerical experiment validation sets and test sets.

**Table 1 sensors-23-08341-t001:** Statistics of clot size and average PC.

Clot Size	Quantity	Average PC
16–20 mm	24	98.24415%
21–26 mm	43	98.57684%
>26 mm	20	98.63558%

**Table 2 sensors-23-08341-t002:** Summary of experiment parameters.

Skull	Data Size	Acoustic velocity
	256 mm × 300 mm × 200 mm	2618 m/s
Array	Number	Center frequency
	512	700 kHz
Clot	Diameter	Velocity
	50 mm	2222 m/s
Excitation	Type	Frequency
	Ricker wavelet	700 kHz

## Data Availability

Some or all data that support the findings of this study are available from the corresponding author upon reasonable request.

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
