# Peer review of "3D Ultrasonic Brain Imaging with Deep Learning Based on Fully Convolutional Networks"

_sensors, 2023, doi:10.3390/s23198341_

Round 1

Reviewer 1 Report

The manuscript entitled “3D Ultrasonic Brain Imaging with Deep Learning Based on Fully Convolutional Networks “has been investigated in detail. The topic addressed in the manuscript is really very interesting and the manuscript contains practical meanings. My recommendation is “Accept”, however a few comments could be consider by authors as follow:

1.      Most of figures are unclear. Increasing the size of the images for better visibility is suggested.

2.      Do the authors check reproducibility for experiments?

3.      How was the best architecture of network obtained? Please discuss it in the manuscript precisely.

Author Response

We would like to thank the reviewer for his/her thoughtful comments and constructive suggestions after thorough reading of our manuscript. We have made the changes stated below to address these points. The overall changes suggested from the reviewers have helped to improve the quality of our manuscript. Please refer to the PDF for a detailed explanation of the specific modifications.

Reviewer 2 Report

Dear authors,

 I think your paper include a very nice method to use deep learning in brain ultrasound via the convolutional networks. The paper is very clear and very good structured. I have just a few comments:

a) In may pictures you include color bars, but the parameter, variable and units that you are reading there is not included. For example in Figure 3 and Figure 6.  It would help the reader if you include the variable and units in  the color bars.

b) In table 2 we see some properties of the transducer that you are using for the measurements with the phantom. However, the electrical impedance (with magnitude and phase) are not included, as well as the beam field of the transducer. This will bring much more information about the characterization of the tranducer, penetration depth and so on.

c) The construction of the phantom used for the in vitro experiments is not explained in detail. It is mentioned that a resin material is used, but not anything more. Could you explain the fabrication of the phantom a bit more in detail? For example how do you form the phantom, its acoustic characteristics, how did you emulate the skull, what is important to take into consideration during the fabrication?

I added some minor things in the attached PDF. You can look into them in my comments.

Thanks for the nice paper.

Author Response

(The authors gave the same response as above.)

Reviewer 3 Report

This study introduces a 3D AI algorithm, 3D BIFCN, combining waveform modeling and deep learning for precise brain ultrasound reconstruction and can accept as a communication paper. Howerver, my comments are as follow:

1-Introduction section is too long and needs to revise, specially by focusing on recent published works.

2- There is no any information about data set and their features, which used in the convolution formula.

3- More explanation about the methods and the number of data are required.

3- Moderate editing of English language and grammar spelling are required.

This manuscript can accept after minor revision as a communication paper.

Moderate editing of English language and grammar spelling are required.

Author Response

(The authors gave the same response as above.)

Round 2

Reviewer 2 Report

Dear Authors,

thank you for answering to my comments and apply the corrections in your paper. I think that now the scientific impact increased, so it can be published as it is. Keep doing such as nice research.

Regards

Belen